# Removal of Polytungstate from Mine Wastewater Using a Flat Renewal Membrane Reactor with N1633 as a Carrier

**DOI:** 10.3390/ijerph191711092

**Published:** 2022-09-05

**Authors:** Liang Pei, Jia Duo, Linlin Chu

**Affiliations:** 1National Engineering Technology Research Center for Desert-Oasis Ecological Construction, Xinjiang Institute of Ecology and Geography, Chinese Academy of Sciences, Urumqi 830011, China or; 2Xinjiang Key Laboratory of Environmental Pollution and Bioremediation, Xinjiang Institute of Ecology and Geography, Chinese Academy of Sciences, Urumqi 830011, China; 3Institute of Geographic Sciences and Natural Resources Research, Chinese Academy of Sciences, Beijing 100101, China; 4College of Agricultural Science and Engineering, Hohai University, Nanjing 211100, China

**Keywords:** simulated mine wastewater, amine extractant N1633, polytungstate, flat renewal membrane reactor

## Abstract

A novel flat renewal membrane reactor (FRMR) with mixed amine extractant N1633 dissolved in kerosene and NaOH solvent was studied for the removal of polytungstate [expressed as W_7_O_24_^6−^ or W (VI)] from simulated mine wastewater. FRMR contains three parts: the feeding cell, reacting cell and renewal cell. A flat membrane of polyacrylonitrile (PAN) was used in the reacting cell, which used mixed solutions of kerosene and NaOH. The amine extractant (N1633) was used as the carrier, and simulated mine wastewater containing polytungstate was used as the feeding solution. The influencing factors of the pH and the other ion strengths in the feeding solutions, the volume ratio of kerosene to NaOH solution, and the N1633 concentration in the renewal solutions were investigated in order to obtain the optimum technique parameters. It was found that the removal percentage of polytungstate could reach 92.5% in 205 min, when the concentration of the carrier (N1633) was 0.18 mol/L, the volume ratio of kerosene and NaOH in the renewal cell was 3:2, the pH in the feeding cell was 4.00, and the initial concentration of polytungstate was 3.50 × 10^−4^ mol/L. The stability and feasibility of FRMR were tested by the investigation of polytungstate retention and the reuse of the membrane.

## 1. Introduction

Tungsten is a rare refractory metal with excellent physical, chemical and mechanical properties. Tungsten and its compound products are widely used in the fields of national economy, national defense construction and high-tech industry [1,2]. They have become indispensable and important raw materials and functional materials in modern society [3,4]. According to the data released by the U.S. Geological Survey in January 2021, the global tungsten and compound resource reserves will be 3.4 million tons in 2020, of which China’s tungsten and compound resource reserves will account for 56%, which is an absolute advantage in the world [5,6].

Wastewater containing tungsten and its compounds mainly comes from mining, beneficiation, metallurgy, electroplating, and glass industries [7,8]. This kind of wastewater is characterized by a low metal ion concentration, and tungsten and compounds belong to the rare and precious metals. If the wastewater containing tungsten and its compounds is discharged directly without treatment, it will not only harm the environment but also waste a lot of resources [9,10,11,12]. At present, a lot of research has been carried out on the treatment methods of wastewater containing tungsten and its compounds. The common main treatment methods include precipitation method, solvent extraction method, ion exchange method and activated carbon adsorption method [5,13]. However, these methods were difficult to recover the tungsten and its compounds resources in wastewater, or the recovery cost was high, and hardly completely extract tungsten and its compounds [14,15]. Liquid membrane separation technology has the advantages of good selectivity, fast extraction speed, simple equipment and energy saving [16,17]. At present, it has been studied and applied in petrochemical industry, hydrometallurgy, medicine, agriculture and wastewater treatment [18,19,20,21]. The removal of tungsten and its compounds by emulsion liquid membrane method has been reported [8,9,10].

In past decades, the removal and reuse of heavy metals from wastewater by liquid membranes (LMs) coupled with variations of carriers (N503, P507 and TBP) were recognized to have broad application prospects, due to their advantages of having a high speed, needing a short time in operation, having a high enrichment ratio, consuming less reagent, and having a low cost [22,23,24,25,26]. Among these technologies, supported liquid membrane (SLM) is widely used, particularly due to its convenience in operation and low costs in surface active agents. However, the instability of SLM remains to be overcome.

In order to solve the problems of the difficult treatment of polytungstate wastewater and the low efficiency of traditional methods, we proposed a new method based on the traditional supported liquid membrane. At the same time, this new method should overcome the problems of membrane stability and durability. In this work, a novel flat renewal membrane reactor (FRMR) with a mixture of N1633 dissolved in kerosene and NaOH solvent was designed based on SLM to efficiently remove W (VI) from mine wastewater with high stability. N1633 was used as a carrier, and kerosene was used as the organic solvent. The influencing factors on the removal of polytungstate and the stability of the novel FRMR were discussed with the expectation of breakthrough industrial applications in the future. As such, there are still some gaps in ore effluent treatment technology compared to the developed country, while this study will supply the gap with the FRMR in the mine wastewater treatment industry.

In brief, the purpose of this study was to solve the problems of polytungstate pollution and its difficult recovery in mine wastewater. This would develop more efficient, more environmental friendly, and more stable polytungstate removal and recovery technology. Simultaneously, it lays a theoretical and scientific foundation for future industrial applications.

## 2. Materials and Methods

### 2.1. The Design of the FRMR and Reaction Mechanisms

The framework of the FRMR and its reaction mechanisms are shown in Figure 1. Basically, the FRMR contained three cells—namely the feeding cell (400 mL), reacting cell (300 mL) and enrichment cell (200 mL)—connected with the feeding pump and renewal pump. The reacting cell was separated into a feeding part and a renewal part by the flat membrane, with the effective volume of each part of the reacting cell being 200 mL. W (VI) and buffer solution (HAC-NaAc) in the feeding cell was poured into the feeding part of the reacting cell by the feeding pump. The certain volume ratio of the mixed oil–water solution in the renewal cell was placed into the renewal part of the reacting cell by the renewal pump. The renewal solution was used to simultaneously remove W (VI) and the renewal membrane so as to simultaneously improve the stability and improve the removal percentage of W (VI). The mixed oil–water solution included membrane solvent (kerosene) with carrier N1633 and stripping solution (NaOH).

The co-removal involves various (equilibrium) reactions: (a) the diffusion of W (VI) from the feeding part into the interface between the membrane and the feeding solution; (b) the extraction of W (VI) from the feeding solution with the carrier N1633 in kerosene, expressed as chemical Equations (I) and (II) in Figure 1, which resulted in the metal complex [([N1633]H)_6_W_7_O_24_]; (c) the diffusion of the metal complex [([N1633]H)_6_W_7_O_24_] through the membrane from the interface between the membrane and the feeding solution of the feeding part to the interface between the membrane and the renewal solution of the renewal part in the reacting cell; (d) the decomplexation of [([N1633]H)_6_W_7_O_24_] into N1633 and W (VI) in the interface between the membrane and renewal solution, according to chemical Equation (III) in Figure 1. NaOH has been used as a stripping agent of W (VI), and kerosene w has been used as the solvent of N1633 in the renewal part of the reacting cell; (e) the enrichment of W (VI) in the renewal cell; (f) the re-input of the N1633 by the renewal pump from the renewal cell into the reacting cell, which could react with the feeding solution to increase the stability of the system, due to the diffusion of the N1633 through the membrane.

### 2.2. Materials and Reagent

A flat membrane of porous polyacrylonitrile (PAN) membrane was used in our new design, with a pore size of 0.26 μm, a thickness of 73 μm, a tortuosity of 1.74, and a porosity of 70~80% (Shanghai Yadong nuclear grade resin Co., Ltd., Shanghai, China). Amine extractant (N1633) was used as the carrier in this work, with a density of 0.871 and a purity of 96% (Shanghai laiyashi Chemical Co., Ltd., Shanghai, China). HCl and W (VI) solution was mixed as a feeding solution to simulate the industrial wastewater containing W (VI). HAc-NaAc buffer solution was used for the pH adjustment (2.4–4.2) of the feeding solution, and the mixed solution of NaCl and KNO_3_ was used for the regulation of the ion strength in the feeding solution to simulate the industrial wastewater. NaOH was selected as the stripping solution, and the self-made kerosene was used as the organic solvent. Mixed solutions of kerosene with N1633 and NaOH solution were used as the renewal solution. All of the reagents (except the kerosene) were of analytical grade.

### 2.3. Test Method

A Digital Acidity Ion Meter (pHS-3C; Shanghai Kangyi Instrument Co., Ltd., Shanghai, China) was used to determine the pH of the solution. The concentration of W (VI) was determined by spectrophotometry with 4-(2-pyridine-azo) resorcinol (PAR) as the chromogenic agent, and the absorbance was measured at 470 nm. The removal efficiency (*R**r*) and separation coefficient were calculated as the removal percentage; they were calculated using Equation (1).
(1)$$Rr=\frac{C_{0}-C_{t}}{C_{0}}\times 100$$

### 2.4. Experimental Procedure

All of the experiments were accomplished at 20 ± 6 °C with the FRMR. The effective area was 40 cm^2^. The flow rates of the two pumps were adjusted to 12.7 mL/min. Before the experiment, the PAN membrane was first immersed into the kerosene solvent with N1633 for an hour, and was then dried naturally and fixed into the reacting cell. The prepared feeding solution and renewal solution were poured into the feeding cell and renewal cell separately. Then, the experiments were started formally with the starting of both feeding pump and the renewal pump. Samples were taken from the feeding cell and the renewal cell for the tests of the W (VI) concentration, at 20, 60, 90, 120, 150, 175, 205 min, respectively.

## 3. Results and Discussion

### 3.1. Effects of pH and Ion Strength in the Feeding Cell

Basically, the concentration difference between the feed part and the renewal part in the reacting cell is the driving power of the mass removing process, which should be impacted by the pH of the feeding solution [12]. In order to investigate the effects of the pH of the feeding solution on the removal efficiency of W (VI) in our tests, the initial experimental conditions were as follows, the ratio of kerosene to NaOH in the renewal cell was stable at 1:1, the concentration of NaOH solution was 0.30 mol/L, and the concentration of carrier was 0.21 mol/L in the mixed solutions of the renewal cell. The initial concentration of W (VI) was adjusted to 3.50 × 10^−4^ mol/L in the feeding cell. As shown in Figure 2, the removal efficiency decreased when the pH of the feeding solution increased from 2.4 to 4.2, and increased with the running time.

Basically, pH is the driving power of the mass diffusion of the complex [([N1633]H)_6_W_7_O_24_], which hypothetically increased the removal efficiency of W (VI) when the pH decreased. Furthermore, when H^+^ increased, this lead to the formation of the ([N1633]H^+^Cl^−^) (Equation (I) in Figure 1), which is of benefit for the generation of [([N1633]H)_6_W_7_O_24_] (Equation (2) in Figure 2). However, in our experiment, we found the formation of WO_2_^2−^ when pH decreased, which hinders the reactions of Equation (II) and lowers the removal efficiency. As a result, the lower the pH is, the more inefficient the removal percentage is. We chose a pH of 4.00 as the optimum pH condition of the feeding part during the following experiments. When the pH was 4.00, the removal percentage of W (VI) was 88.9%.

In the industrial wastewater, there always existed other ions, which may affect the removal of W (VI). Thus, the NaCl and KNO_3_ were used to simulate the industrial wastewater, and to investigate the effects of the initial ion strength in the feeding cell on the removal percentage of W (VI). As shown in Figure 3, the removal percentage of W (VI) increased when the initial ion strength changed from 0.4 mol/L to 1.7 mol/L, and was maintained higher than 80%.

### 3.2. Effects of the Volume Ratio and Carrier Concentration in the Renewal Cell

As we described above, the mixed solutions in the renewal cell were a certain volume ratio of kerosene with N1633 to NaOH. In order to investigate the volume ratio effects of the mixed solutions, the ratios of kerosene to NaOH were adjusted to 1:5, 1:2, 1:1, 3:2, 4:3, 5:2, which is shown in Figure 4. The result suggested that the removal efficiency of W (VI) first increased with the volume ratio of kerosene to NaOH, when it was lower than 3:2. However, the removal efficiency of W (VI) decreased sharply when the volume ratio of kerosene to NaOH reached 3:2.

In our work, kerosene with N1633 was selected as the membrane solution, which could recover the stability of the membrane in order to improve the complexing percentage to increase the efficiency of W (VI) removal, by increasing the chances of the recycling of the N1633. NaOH was selected as the stripping solution for the removal of W (VI), the increase of which not only increased the stripping percentage for the decomplexation of [([N1633]H)_6_W_7_O_24_] but also increased the concentration differences of the H+ concentration in the feeding part and renewal part of the reacting cell. Finally, the diffusion of both [([N1633]H)_6_W_7_O_24_] and the recycled N1633 were increased. When the volume ratio of the mixed solutions increased, the complexing rate increased while the stripping rate and diffusion rate decreased. The balance of these reactions resulted in the maximum removal efficiency of W (VI), when the volume ratio of kerosene to NaOH was 3:2. Meanwhile, the final results showed that when the volume ratio was 3:2, the removal percentage of W (VI) was 89.7%.

Moreover, N1633 played important roles in the removal of the W (VI). Based on Equations (I) and (II) in Figure 1, the higher the N1633 is, the higher the chances of the formation of complexation [([N1633]H)_6_W_7_O_24_] are. The higher probability of the diffusion and decomplexation of the [([N1633]H)_6_W_7_O_24_] increased the removal efficiency, as shown in Figure 5. Overall, considering of the removal efficiency and solvent cost, 0.18 mol/L was selected as the optimum carrier concentration. The removal percentage of W (VI) was 92.5%.

### 3.3. Effects of W (VI) Retention and the Reuse of the Membrane

In the previous investigations of the removal of heavy metal using SLM, the retention of the heavy metal ion on the membrane was observed and concerned [12,13]. In this work, the W (VI) ion was also concerned. According to the concentration of W (VI) in both the feed cell and renewal cell, the concentration of W (VI) on the membrane can be calculated. As shown in Figure 6, the detention of W (VI) on the membrane increased with the running time. However, the percentage decreased with the running time, which resulted in the W (VI) percentage on the membrane remaining stable when the running time is longer than 205 min.

This is due to the decrease of the decomplexation percentage of [([N1633]H)_6_W_7_O_24_] in the interface between the membrane and renewal solution with the increase of W (VI) in the renewal solution of the reacting part when the running time increased. With the increase of the running time, the balance reached and the detention of W (VI) no longer increased, with approximately 16% of the W (VI) on the membrane of the reacting cell. However, this did not have negative effects on the renewal of the membrane. As shown in Figure 7, the removal percentage remained higher than 80% when the experiment was reused six times, which verified the stability and feasibility of the FRMR. The stability of the membrane and the removal percentage of W (VI) was also enhanced by the separation of the reacting cell from the feeding and renewal cells in the novel design. This is precisely the advantage of our newly designed FRMR, which avoids the falling off of the carrier that may be caused by the stirrer in the traditional SLM method.

## 4. Conclusions

In view of the difficulty of removing polytungstate from mine wastewater, a flat renewal membrane reactor (FRMR) was studied. The running and the influencing factors of this FRMR with a mixture of N1633 dissolved in kerosene and NaOH solvent for the removal of the polytungstate from the simulated wastewater was considered in the laboratory in this work. The results showed that FRMR is able to remove polytungstate from simulated industrial wastewater with high removal efficiency, using the mixture of a carrier of N1633 and NaOH (stripping solution) as a renewal solution. The optimum technology parameters were obtained based on the effects of different influencing factors, with consideration for both the removal efficiency and solvent costs. The resulting optimum parameters of the FRMR were: the concentration of the carrier (N1633) was 0.18 mol/L, the volume ratio of kerosene and NaOH in the renewal cell was 3:2, the pH in the feeding cell was 4.00, and the running time was 205 min. Under the optimum conditions, the removal percentage was 92.5%. Although there is a detainment of polytungstate on the membrane during the operation, the stable retention percentage remained lower than 18%. When the membrane was used six times, the removal rate of polytungstate was still higher than 80%. However, in order to apply the FRMR on a larger scale, the stability and durability of the membrane modules should be solved further in the future.

## Figures and Tables

**Figure 1 ijerph-19-11092-f001:**
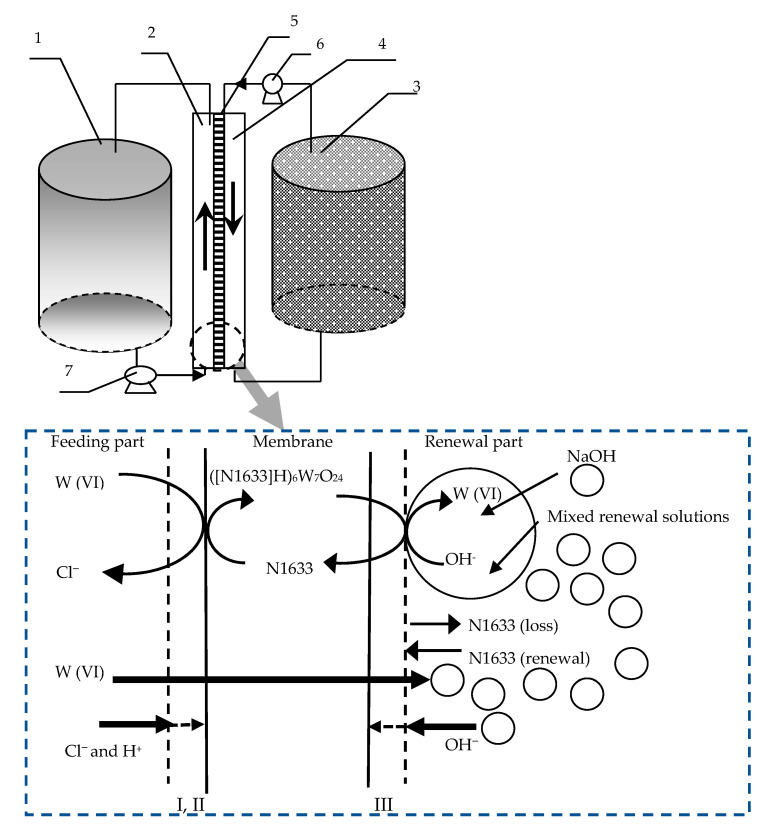
The structure and reaction mechanisms of the flat renewal membrane reactor (FRMR) with a mixture of N1633 dissolved in kerosene and NaOH solvent.

**Figure 2 ijerph-19-11092-f002:**
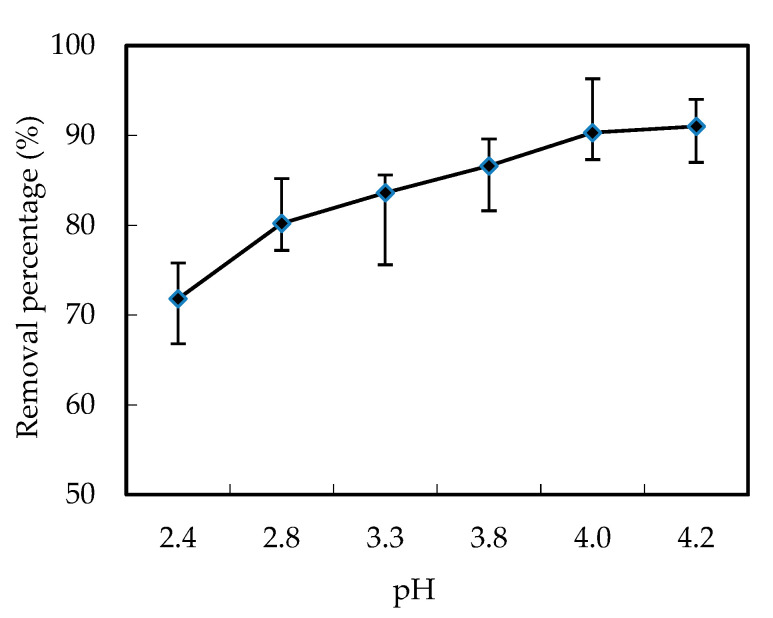
Effects of the pH of the feeding solution on the removal efficiency of W (VI).

**Figure 3 ijerph-19-11092-f003:**
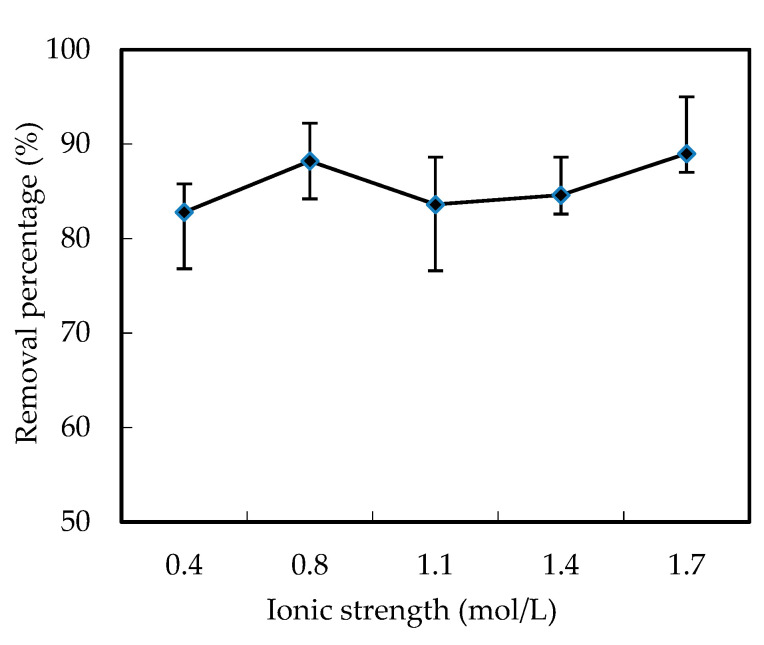
Effects of the ion strength in the feeding cell on the removal percentage of W (VI).

**Figure 4 ijerph-19-11092-f004:**
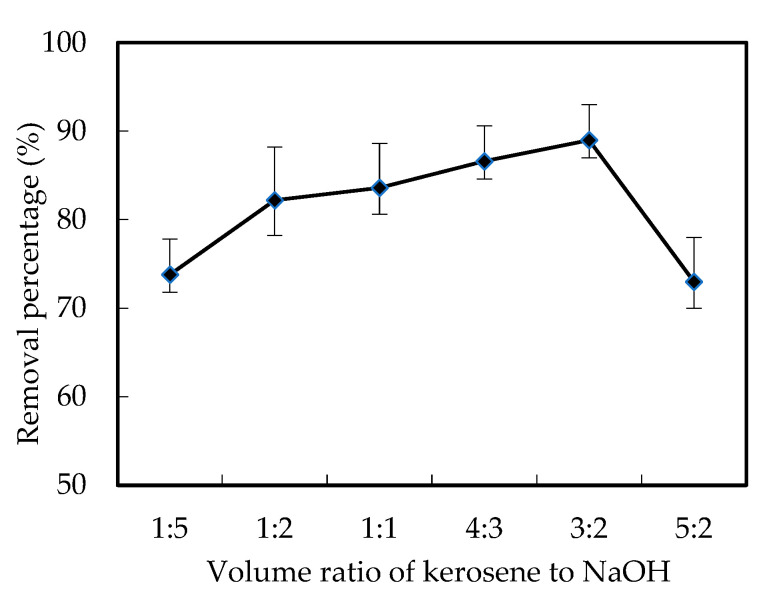
Effects of the volume ratio of kerosene to NaOH on the removal efficiency of W (VI).

**Figure 5 ijerph-19-11092-f005:**
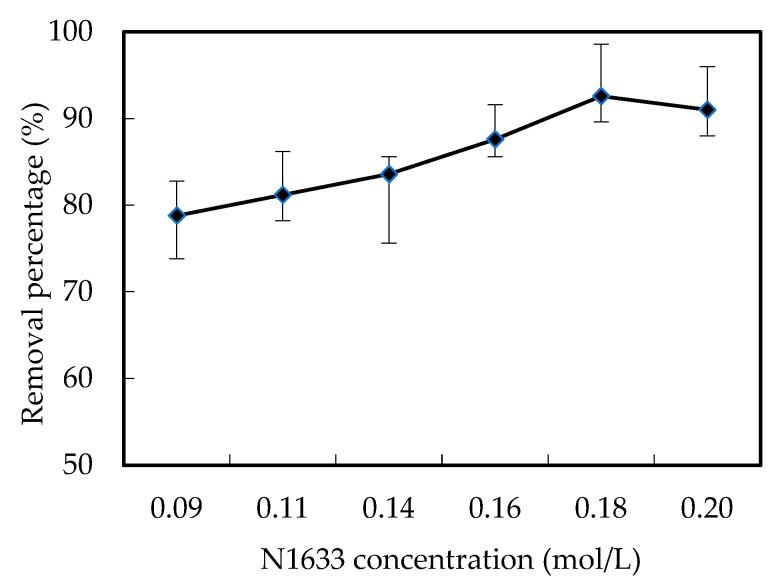
Effects of the N1633 concentration on the removal percentage of W (VI).

**Figure 6 ijerph-19-11092-f006:**
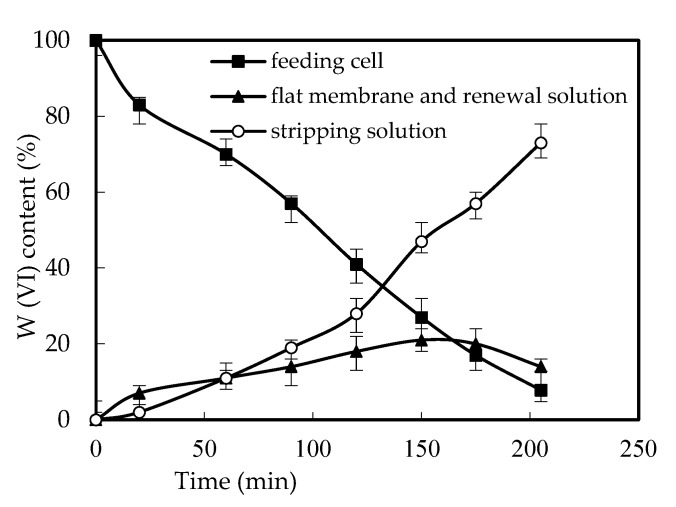
The retention of W (VI).

**Figure 7 ijerph-19-11092-f007:**
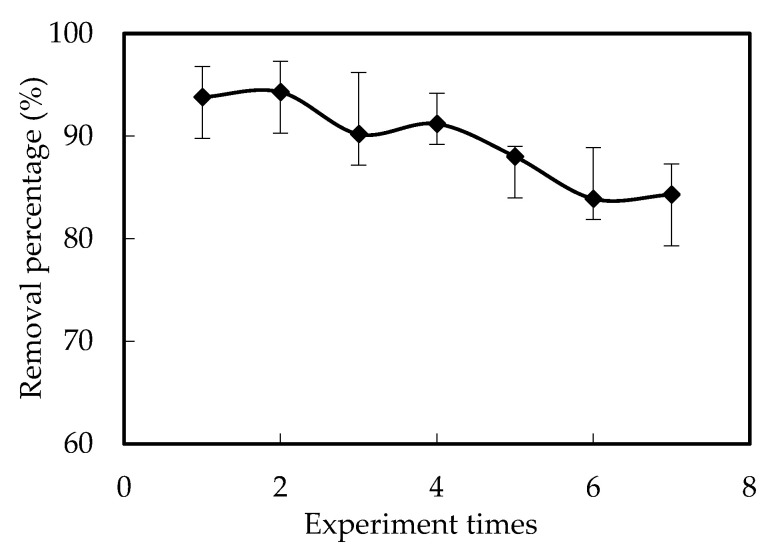
The effects of the reuse time on the removal of W (VI).

## Data Availability

This study does not report any data.

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
