# Peer review of "Removal of Polytungstate from Mine Wastewater Using a Flat Renewal Membrane Reactor with N1633 as a Carrier"

_ijerph, 2022, doi:10.3390/ijerph191711092_

Round 1

Reviewer 1 Report

The conclusions must be properly written and outline the competitive developments. The extent to which the posed problem is solved and its perspectives shall be outlined.

The data provided in Fig. 4 must be critically described. The sources of the presented inaccuracies must be shown.

In Fig. 6, error bars must be provided.

Fig. 7 does not have a clear caption. Furthermore, the quality of the figure is low.

The relevant source must be cited.

Selective chemical binding enhances cesium tolerance in plants through inhibition of cesium uptake // Scientific Reports 5 (1), 1-10, 2015

'the retention of the membrane should be further tackled in the 229 future for the application of the FRMR in larger scale.'

The sentence is grammatically flawed.

'Under the optimum conditions, the removal rate could reach 225 92.5%.'

Use percentage instead of rate and define the units properly.

Fig. 1 does not provide a clear introduction to the work presented.

Author Response

Responses to Reviewers

1 Comments and Suggestions for Authors

1.The conclusions must be properly written and outline the competitive developments. The extent to which the posed problem is solved and its perspectives shall be outlined.

Response: Thank you for your suggestion. We rewrote the conclusion and displayed it in red fonts

2.The data provided in Fig. 4 must be critically described. The sources of the presented inaccuracies must be shown.

Response: Thank you for your advice. We changed the figure to percentage as ordinate, and included the error bar. In this way, we can see the law more clearly.

3.In Fig. 6, error bars must be provided.

Response: Thank you for your advice. We add error bar in these figures.

4.Fig. 7 does not have a clear caption. Furthermore, the quality of the figure is low.

Response: Thank you for your advice.We have changed this.

5.The relevant source must be cited.

Selective chemical binding enhances cesium tolerance in plants through inhibition of cesium uptake // Scientific Reports 5 (1), 1-10, 2015

Response: Thank you for your advice.We have added this.

  1. 'the retention of the membrane should be further tackled in the 229 future for the application of the FRMR in larger scale.'

Response: Thank you for your advice.We have modified this with red fonts.

7.The sentence is grammatically flawed.

'Under the optimum conditions, the removal rate could reach 225 92.5%.'

Response: Thank you for your advice.We have modified this with red fonts.

  1. Use percentage instead of rate and define the units properly.

Response: Thank you for your advice.We have modified to percentage insteading of rate in all text.

  1. Fig. 1 does not provide a clear introduction to the work presented.

Response: Thank you for your advice.We have changed it with red fonts and improved Figure 1 .Please see the new figure1.

Reviewer 2 Report

Manuscript:  IJERPH-1835373 _reviewer comments

Title: Removal Wolfram(VI) from the Mine Wastewater Using Flat Renewal Membrane Reactor with N503 as Carrier

Recommendation: major revisions with a resubmission

            The manuscript was submitted for possible publication. The manuscript reports on the variation in the DOC of soil.

The present version requires extra edits which include more formatting, and polishing. Conclusions are very lengthy and confusing to elucidate the findings and highlight the relevance and the novelty of the work and finally get a clear contribution of the study towards further research and applications.

The general comments are to be addressed before consideration for publication.

1.      I feel that further English polishing is essential to considerably improve the manuscript.

2.      The manuscript significantly requires structuring and formatting to better reflect the content of the investigations. The manuscript lack cohesion, such as very lengthy text in most of the manuscript, and a good linkage between ideas is quite disjointed the reader from the article.

The specific comments for sections.

Introduction and Experimental section:

·         Generally, the M and M have details that are supported by past studies and sufficient to refer to literature or reason for selection

·         There is no statistical analysis provided in Materials and Methods or results and discussion to confirm the significant differences, The text did not specify how many tests were done for normality and null hypothesis and samples and # of experiments duplication with no error bars.

·         L505: I cannot reconcile how the entire work excluded the sample preparation requirement's effect

·         L76-79: “The certain volume ratio of the mixed oil‐water solution, including membrane solvent (kerosene) with carrier N503 and stripping solution (NaOH), in the renewal cell was placed into the renewal part of the reacting cell by the renewal pump for the removal of W (VI) and renewal of the membrane simultaneously, as to improve the stability and increase the removal rate of W (VI) at the same time.” Very lengthy sentence and the key message of the content is completely unclear to me. Also, what is the need for certain volume ratio

Results and Discussions and Conclusions:

·         L134-136: “The initial experimental conditions were that: ratio of kerosene to NaOH in the renewal cell at constant at 1:1, the concentration of NaOH solution at 0.30 mol/L, the concentration of carrier at 0.21 mol/L in the mixed solutions of the renewal cell.” How do these concentrations match the 1:1 ratio and the reason for this specific ratio, is it a molar ratio?

·         The results have major limitations without considering the variation of temperature, the authors should highlight and verify their rationale and concept with 20

·         Please highlight limitations of the study

·         The authors would need to clearly highlight the novelty of the work and the gaps addressed in comparison to previous studies.

·         L143-148: very lengthy and the key message is completely confusing with regards to pH effect in the current study with respect to the regular trend of removal. It is suggested to add more comparison and clear structured information and justification. Please refer to more details about Figure 2. Was there any statistical analysis done?

·         The general organization of the manuscript is poor, especially in the results and discussion section

·          Conclusion is not robust enough

·         Under the optimum conditions, the removal rate could reach 92.5%”. What rate the authors refer to that has % and no unit. The removal efficiencies and parameters should be presented in a table format for all conditions 

Author Response

Manuscript:  IJERPH-1835373 _reviewer comments

Title: Removal Hexavalent Tungsten from the Mine Wastewater Using Flat Renewal Membrane Reactor with N1633 as Carrier

Recommendation: major revisions with a resubmission

      The manuscript was submitted for possible publication. The manuscript reports on the variation in the DOC of soil.

The present version requires extra edits which include more formatting, and polishing. Conclusions are very lengthy and confusing to elucidate the findings and highlight the relevance and the novelty of the work and finally get a clear contribution of the study towards further research and applications.

The general comments are to be addressed before consideration for publication.

Response: Thank you for your advice. We have added an innovative explanation to the introduction and conclusion. Please look at the red font.

  1. I feel that further English polishing is essential to considerably improve the manuscript.

Response: Thank you for your advice. We have made some changes to English. Please see the red font.

  1. The manuscript significantly requires structuring and formatting to better reflect the content of the investigations. The manuscript lack cohesion, such as very lengthy text in most of the manuscript, and a good linkage between ideas is quite disjointed the reader from the article.

Response: Thank you for your advice. We added the sentence of paragraph link and modified the expression. Please see the red font

  1. The specific comments for sections.

3.1 Introduction and Experimental section:

  • Generally, the M and M have details that are supported by past studies and sufficient to refer to literature or reason for selection
  • There is no statistical analysis provided in Materials and Methods or results and discussion to confirm the significant differences, The text did not specify how many tests were done for normality and null hypothesis and samples and # of experiments duplication with no error bars.

Response: Thank you for your advice. We added the error data bar in all text.Please the new figures1-7.

  • L505: I cannot reconcile how the entire work excluded the sample preparation requirement's effect.

Response: Thank you for your advice. We added relevant literature to explain.

  • L76-79: “The certain volume ratio of the mixed oilwater solution, including membrane solvent (kerosene) with carrier N503 and stripping solution (NaOH), in the renewal cell was placed into the renewal part of the reacting cell by the renewal pump for the removal of W (VI) and renewal of the membrane simultaneously, as to improve the stability and increase the removal rate of W (VI) at the same time.” Very lengthy sentence and the key message of the content is completely unclear to me. Also, what is the need for certain volume ratio

Response: Thank you for your advice. We splited it into two sentences. As:The certain volume ratio of the mixed oil-water solution in the renewal cell was placed into the renewal part of the reacting cell by the renewal pump for the removal of W (VI) and renewal of the membrane simultaneously, as to improve the stability and increase the removal percentage of W (VI) at the same time. The mixed oil-water solution included membrane solvent (kerosene) with carrier N1633 and stripping solution (NaOH). Part3.2 ,the volume ratio of kerosene to NaOH set to 1:5, 1:2, 1:1, 3:2, 4:3, 5:2, separatly.

3.2 Results and Discussions and Conclusions:

  • L134-136: “The initial experimental conditions were that: ratio of kerosene to NaOH in the renewal cell at constant at 1:1, the concentration of NaOH solution at 0.30 mol/L, the concentration of carrier at 0.21 mol/L in the mixed solutions of the renewal cell.” How do these concentrations match the 1:1 ratio and the reason for this specific ratio, is it a molar ratio?

Response: Thank you for your advice. We first prepare the solution with the required molar ratio, and then mix it according to the volume ratio.

  • The results have major limitations without considering the variation of temperature, the authors should highlight and verify their rationale and concept with 20

Response: Thank you for your advice. In future research, we will increase the research of temperature. However, some literatures and references show that temperature has less effect on the system than other factors.

  • Please highlight limitations of the study. The authors would need to clearly highlight the novelty of the work and the gaps addressed in comparison to previous studies.

Response: Thank you for your advice. We have added some explanations about limitations and novelty in the introduction and conclusion. Please see the red font.

  •  
  • L143-148: very lengthy and the key message is completely confusing with regards to pH effect in the current study with respect to the regular trend of removal. It is suggested to add more comparison and clear structured information and justification. Please refer to more details about Figure 2. Was there any statistical analysis done?

Response: Thank you for your advice. We have done repeated experiments, but there is no mark. We now add error bars. Please see the charts of the full text.

  • The general organization of the manuscript is poor, especially in the results and discussion section

Response: Thank you for your advice. We have made some changes. Please see the red font.

3.3 ·          Conclusion is not robust enough

  • Under the optimum conditions, the removal rate could reach 92.5%”. What rate the authors refer to that has % and no unit. The removal efficiencies and parameters should be presented in a table format for all conditions 

Response: Thank you for your advice. We have unified the unit, which is expressed in%. All the pictures have been redrawn. In the conclusion, we also summarize the optimum conditions. And we added error bars, so that we can see the reliability of the conclusion,basically. There are still some problems in the research. In the conclusion, we also explain the direction of future efforts. We will gradually improve these experiments.

Reviewer 3 Report

1. One of the major concerns is that there is no evidence or data to show the reproducibility of data in figures. It seems all experiments have been done once. If yes, the data can not be justified. if repeated, error bar for all data should be drawn into Figures.

2. Fig. 7: Just explained the removal capability is more than 80%, without elaboration the observed fluctuations in data. Need to be elaborated such changes.

Author Response

3  Comments and Suggestions for Authors

  1. One of the major concerns is that there is no evidence or data to show the reproducibility of data in figures. It seems all experiments have been done once. If yes, the data can not be justified. if repeated, error bar for all data should be drawn into Figures.

Response: Thank you for your advice. We have done repeated experiments, but there is no mark. We now added error bars. Please see the any chapters and charts of the full text.

  1. Fig. 7: Just explained the removal capability is more than 80%, without elaboration the observed fluctuations in data. Need to be elaborated such changes.

Response: Thank you for your advice. And, We redrawn figure 1 ,figure 7 and added error bars. We can probably see the law. There are still some substances left in the membrane pores and on the membrane surface. At present, we have no conditions for determination, and we can only calculate them through existing figures.

Round 2

Reviewer 1 Report

The authors did not fix the issues raised by the reviewers.

The error bars look unrealistic.

"In the future, we will 267 conduct some application studies with business to explore their practical application ef‐ 268 fects"

Useless statements.

The graphic is of poor quality.

Author Response

Responses to Reviewer 1

Comments and Suggestions for Authors

The authors did not fix the issues raised by the reviewers.

 Response: Thank you for your suggestion. In the first revision, we forgot to replace the literature you suggested. Now it has been replaced. Some changes have also been made in the conclusion, see red font. Figs. 1 and 6 are also redrawn and replaced.

The error bars look unrealistic.

 Response: Thank you for your advice. We redraw the figure6. Last time, one line of the error bar in Fig. 6 input data incorrectly. Now we have corrected it. In addition, please note that the vertical coordinate range of each drawing is different, some are 60-90 and some are 0-100. Therefore, the length of error bars is different.

"In the future, we will 267 conduct some application studies with business to explore their practical application ef‐ 268 fects"

 Useless statements.

 Response: Thank you for your advice. We modified some statements. This redundant sentence has also been deleted.

The graphic is of poor quality.

Response: Thank you for your suggestion. We redraw and replace all the graphs. Please guide again.

Reviewer 3 Report

Accept

Author Response

Responses to Reviewer 3

Comments and Suggestions for Authors

Accept

Response: Thank you for your affirmation of my article.Best wishes.